# XAI PROCEDURAL FAIRNESS AUDITING: AVOID MISGUIDED OUTCOMES BY REFOCUSING ON FAIRNESS PROPERTIES

## ABSTRACT

Interpretable machine learning and explainable AI (XAI) methods used to investigate fairness properties can be described as ML auditing. Current ML researchers have noted that there are limited, successful implementations of procedural fairness, which focuses on the decision-making steps rather than fair outcomes. We present the results of our **procedural fairness auditing framework** for XAI tools. We evaluated Stealth, an ensemble XAI method that combines novel global surrogate model generation that avoids detection by deceptive models with well-known LIME's local explanations. Through a procedural fairness lens, we audited Stealth's decision-making process outside of its notable performance outcomes. The procedural fairness audit reports that Stealth's global surrogate models are impressive and a successful application of recursive bi-clustering for representative data downsampling. However, the audit also revealed Stealth's training data biases, and we discuss how Stealth's fairness claims were misguided by "fairer outcomes." The procedural fairness auditing framework provides an outline of how to interpret ML decision-making, ensuring procedural fairness.

## 1 INTRODUCTION

Researchers have identified a need for interpretable machine learning (IML) and explainable AI (XAI) methods that explain models' decision-making processes, including fairness and bias properties. These research interests are combined under the *ML auditing* domain. IML and XAI methods have been extensively studied, and some, such as Local Interpretable Model-Agnostic Explanations (LIME) (Ribeiro et al., 2016) and SHapley Additive exPlanations (SHAP) (Lundberg & Lee, 2017) have become very popular. Recent studies (Slack et al., 2020) fooling LIME and SHAP have highlighted deceptive, malicious AI and the dangers of AI detecting when it is being explained or *audited*. However, there is little known about auditing a potentially deceptive agent. The concepts of deceptive AI and ML auditing are central to understanding black box models and ensuring trustworthy AI.

While various fairness metrics scores can be used to reflect the extent of fairness, another type of fairness testing is to generate *explanations* on what features are *interpreted* to contribute most to a model's outcome. The advantage of using explanation-based testing over statistical metrics is that it provides stakeholders with more straightforward and human-comprehensible insights on how much a feature has influenced the prediction model outcomes. *Procedural fairness auditing* involves interpretation and explanation while emphasizing procedural fairness. Procedural fairness calls for ensuring models' decision-making processes are iteratively fair. Current research interests have progressed from measuring fairness metrics to prioritizing procedural fairness (also known as procedural justice) over distributive fairness, shifting from "fair outcomes" to "fair procedures." Carey & Wu (2023) have critiqued the reductive ways that ML practitioners claim to implement procedural fairness due to a lack of auditing model's outcomes as well as decision-making *process*.

In this paper, we applied our *procedural fairness audit framework* to understand the decision-making process of an explainable AI (XAI) tool **Stealth** (Alvarez & Menzies, 2023). Stealth is an ensemble auditing tool that generates explanations through global surrogates and local explanations with the well-known XAI tool, LIME. Since we chose to investigate an XAI tool, Stealth, our analysis

can be seen from a new *meta-auditing* perspective. There are three layers of explanation within our meta-audit (see Figure 1): (1) the XAI audit system Stealth, which generates global surrogate models; (2) LIME, which provides local explanations for the parent and surrogate models; and (3) a "auditor-in-the-loop" procedural fairness audit which iteratively investigates the case study's algorithm/decision-making process. There is Stealth's auditing explanation process through its surrogate generation and our *meta-audit* of how Stealth generates its explanations. Our iterative audit exposed the latent bias present in Stealth's surrogate generation and further supports why procedural fairness needs to be accurately implemented in socio-technical work. We provide a novel application of procedural fairness through a auditor-in-the-loop auditing framework, and our results clarify the appropriate use of Stealth's deception-aware XAI system. While Stealth's "fairness" claims are misguided, we discuss the impressive contributions of Alvarez & Menzies (2023) surrogate generation technique and how our audit results contribute to the larger Trustworthy ML, XAI, IML, and security research communities.

## 2 Literature Review

ML auditing is a combination of interpretation (how AI makes decisions) and explanation (how these interpretations are explained to humans). *Procedural fairness auditing* investigates a model's decision-making process, prioritizing fairness at each step. Fairness testing refers to the testing schema that evaluates whether bias is identified and mitigated in ML models. This section reviews the current ML Auditing field. However, since there is a gap in specific auditing research, the most relevant work is from the Explainable AI/ML (XAI), Interpretable ML (IML), and Trustworthy ML research domains. We introduce the field of XAI research, the relevant IML and Trustworthy ML work, and discuss the potential challenges faced by current XAI techniques as related to *procedural fairness auditing*.

**Procedural Fairness:** Researchers (Carey & Wu, 2023; Grgić-Hlača et al., 2018; Morse et al., 2021) note procedural fairness (also referenced as procedural justice) would investigate the models' entire ecosystem: the model itself, the data sets used in training, the developer, and the population impacted by the model. Rueda et al. (2022) argue that explainability is necessary to ensure procedural fairness requirements. There are differing opinions about where fairness relates to interpretability and explainability, but for the sake of argument, we view the explanation as a subdomain of interpretability and interpretable ML as directly connected to trustworthy ML. Rudin (Rudin, 2019; Rudin & Radin, 2019; Semenova et al., 2022) has a long-term claim that deep learning and overly complex models have been used simply for perceived sophistication when simpler models perform similarly and competitively with larger models. With a lot of attention on deep learning and computationally heavy, black-box models, strengthening the generalizability of conceptually simpler but still strong models needs more attention.

**XAI Scope:** The standard ways to evaluate the performance of these models vary on their scope: *pre-hoc, in-model, post-hoc, intrinsic, model-specific, model agnostic, local, and global* (Molnar, 2022; Carvalho et al., 2019). Since procedural fairness should be considered at all levels of product development, the guiding questions should be considered before (pre-hoc) training and after training (post-hoc) the model. The procedural fairness auditing framework is a way for practitioners to understand a high-level model agnostic overview of the model through a step-by-step procedural investigation of the entire development pipeline. Our procedural fairness audit falls within the pre-hoc and post-hoc, model-agnostic, global, and local explainability scope.

Global methods will focus on identifying features that most influence the model's outcomes, whereas local methods will identify features that specifically impact individual outcomes as related to an individual input. For example, LIME creates thousands of test set data perturbations. If there are 100 testing points, then for each one, LIME creates a neighborhood of perturbed points around the test point. Each of these perturbation points has very slight changes, and all of these points are given to the model to label. Based on how the labels change, LIME is able to use its perturbed points to identify which feature perturbation impacted the label for that testing point.

We chose Stealth as our case study because its combination of global and local explanations widens our scope beyond just one or the other. Stealth's global surrogate generation falls specifically within the *static, model-agnostic, global, post-hoc, surrogate, feature importance* explanation domain (also referred to as knowledge distillation) (Zhou et al., 2021). Stealth's application of LIME connects

knowledge distillation with a saliency method within the specific *static, model-agnostic, post-hoc, local, feature importance* explanation domain (Zhou et al., 2021). Calls within the IML field (Carvalho et al., 2019; Molnar et al., 2021; Molnar, 2022; Molnar et al., 2020; Kim et al., 2016) and XAI fields (London, 2019; Rueda et al., 2022; Vilone & Longo, 2020; Adadi & Berrada, 2018) showed there were limited explanation systems that were deception aware, and there still are very few defenses against deceptive AI.

**Malicious, Deceptive AI Fooling XAI:** Trustworthy ML and XAI tools have to expand beyond fairness metric performance because researchers have proved simple models like random forests, SVMs, and logistical regression can be maliciously misleading. Slack et al. (2020) explanation detection with SHAP and LIME exposed the vulnerabilities of perturbation-based saliency explanation methods. There has been a response to improving their vulnerabilities, like making them more robust or changing the distribution technique that generates their perturbed data (Molnar et al., 2020; Saito et al., 2020). However, these improvements do not guarantee deception prevention.

**Relevant Auditing Work:** The most related work is by Park et al. (2022). They proposed an auditing tool that separates itself from other fairness toolkits (Bellamy et al., 2018; Bird et al., 2020; Saleiro et al., 2018; Kearns et al., 2018) because it is applicable to cloud services like Stealth. However, their main contributions are providing security for both parties, but their limitations include not building fair ML models nor addressing possible attacks such as model extraction, inversion, and evasion attacks (such as deception). Medical and Biology ML scholars (Eid et al., 2021;?; Oala et al., 2020) have identified a gap in IML research and a need for clear and generalizable auditing methods that are easily tailored to their applications. Specifically, Oala et al. (2020) reported there was a need for audit applications for the entire development life cycle. Amongst the XAI domain, there is a theoretical call for understanding explanations better tailored for humans (Miller, 2019) and XAI auditing tools that consider algorithmic fairness (Adler et al., 2018). Our work differs because our procedural fairness audit presents an "auditor-in-the-loop" (Zhang et al., 2022; Miller, 2019; Abdul et al., 2018; Zhu et al., 2018; Mohseni et al., 2021) auditing framework for developers, and our case study is an XAI tool that is deception-aware.

**Other Fairness & Bias Mitigation Techniques:** Other techniques include pre-processing data interventions (Calmon et al., 2017; Feldman et al., 2015; Chakraborty et al., 2021), reweighting (Kamiran & Calders, 2012), manipulating fairer model outcomes (Hardt et al., 2016; Pleiss et al., 2017; Zafar et al., 2017; Kamishima et al., 2012), trained model mitigation (Zafar et al., 2019; Zhang et al., 2018), and other ways to audit (Kearns et al., 2018; Saleiro et al., 2018).

There is a common theme within the Software Engineering and IML fields to remove demographic features such as race and gender and then imply the model cannot be discriminatory because these individual demographic features are removed from training. This is a very naive understanding of discrimination and connects to the sociological concept of "colorblind racism" Carr (1997). Simply removing demographic features does not change any underlying bias that is still present and can be inferred through other features.

## 3 CASE STUDY MOTIVATION

In the past few years, scholars have raised concerns about deceptive AI (Schneider et al., 2020; Brundage et al., 2018; Banovic et al., 2023; Slack et al., 2020), discriminatory AI (Noble, 2018; Benjamin, 2020; Buolamwini & Gebru, 2018; Keyes, 2018; O'neil, 2017; Obermeyer et al., 2019; Angwin et al., 2016; Eubanks, 2018), and vulnerabilities of XAI tools (Meske et al., 2022; Baniecki & Biecek, 2023; Slack et al., 2020). Thus, more current research has begun to explore paths for better identifying these problems. Researchers have presented many local mitigation, deception, and detection solutions which Schneider et al. (2020) described as *"a new cat and mouse game between 'liars' and 'detectors'... in the context of AI"*, with limited scalable industry applications. Stealth was motivated by a real-world event: Dieselgate (Ewing, 2020; Fracarolli Nunes & Lee Park, 2016), where the U.S. Environmental Protection Agency (EPA) found Volkswagen cars' AI detected when it was evaluated and released wrongfully lower emission rate only when evaluated. There is an assumption that biased, discriminatory AI is an unforeseen side effect, an unfortunate consequence of biased data, or an illuminating reflection of unconscious bias. Still, even hypothetically, we must consider: what if Volkswagen wanted to deceive its EPA evaluators? How will practitioners be able to evaluate larger corporations' pay-for-service models, and how will they know they are not being

deceived? Stealth raised a concern regarding deceptive, malicious AI that has yet to be explored: **How can XAI tools explain models that can potentially detect and mislead explanation tools?**

The novelty of Stealth is applying a well-established downsampling technique, FastMap, to select a subset of training data for surrogate model generation. This technique, known as recursive bi-clustering, is not a novel concept in itself, but the novelty of Stealth lies in its application of bi-clustering to the XAI fairness problem. Alvarez & Menzies (2023) proposed a global surrogate modeling technique as a way to audit deceptive models. It claimed to create accurate and "fairer" global surrogate models that required a remarkably minimal amount of training data.

Researcher Molnar (2023) has recently raised criticisms of the popular use of the Synthetic Minority Over-sampling Technique (SMOTE) (Chawla et al., 2002). Molnar claims that "fixing" an imbalanced data set is misguided if state-of-the-art learners are implemented, and calibration is a concern (Molnar, 2023). Where over-sampling creates generated examples from the original data, downsampling reduces the larger data population to the size of the smallest population. He comments against downsampling as a substitute for over-sampling because of the "loss of data" and the loss of the representative sample. However, Stealth creates a downsampled subset with the same representation balance as the original set. Stealth is a good relevant case study because Stealth's results demonstrate the unique advantage of *representative downsampling* and how limited data can provide insights on how data imbalance impacts bias and fairness properties.

## 4 PROCEDURAL FAIRNESS AUDIT

Our audit uses a top-down research design, the current approach favored by responsible AI researchers (Bringas Colmenarejo et al., 2022; Varshney, 2022; Camacho Ibáñez & Villas Olmeda, 2021). The top-down/proactive approach refers to gathering the interests of stakeholders and investments and deciding the best practices, and the bottom-up/reactive approach is where operational practices inspire best practices after development. The literature supports the top-down proactive approach to organizing high-risk projects. However, a big problem we highlight is the interest of stakeholders and the possibility of deception. Therefore, our audit must be implemented at the start (top-down) and end (bottom-up) of the design pipeline. We guided each step of the procedural fairness audit with leading questions (adapted from the necessary questions in Zhang et al. (2022)), and this section describes the results of our audit.

Once the high-level, top-down understanding was gathered, we moved on to the bottom-up approach, moving through each individual pipeline step outlined in Figure 2. Since Stealth's surrogate generation is rooted in data down-sampling, the most important step was understanding the data sets, data processing, and training sets.

1. What is the motivation behind the socio-technical system, and why should this system be implemented?

2. What are the worst-case impacts of this system, and who is impacted most by the socio-technical system? Are they consulted in the making of this system?

3. How is fairness considered in this context, and where are the areas for possible bias?

4. How is the system intended to work?

5. What data is used to train the model? What are the proportions of protected attributes within the training and test sets?

6. What algorithms, toolkits, or packages are used?

7. What is the overall performance of the model?

8. How is success defined, measured, and evaluated?

9. Is this model intrinsically interpretable? If not, justify why an IML model is not used.

10. How does the model decide globally and locally for a particular instance (e.g., a transaction and an audit engagement)?

11. What feature(s) most informs the trained model's decision-making?

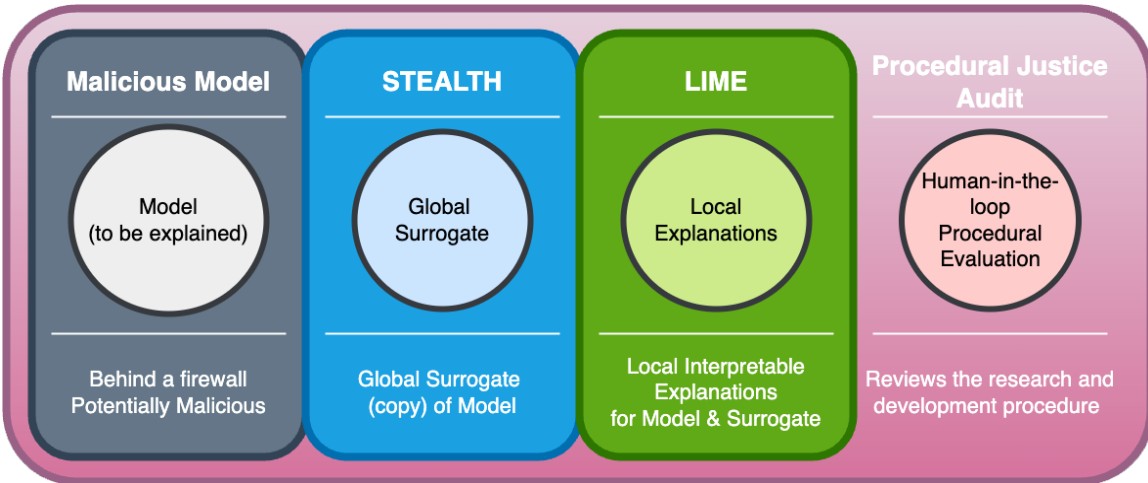

Figure 1: Procedural Fairness Audit.

# 5 CASE STUDY: STEALTH

## 5.1 PRE-PROCESSING

The data sets in Table 1 are popular in AI/ML fairness literature, particularly for their clearly biased training results. Stealth's pre-processing steps were supported by pre-processing steps in IBM's fairness toolkit (Bellamy et al., 2018). For each data set, the standard ML pre-processing methods were applied: (1) rows with missing values were ignored, (2) continuous features were discretized and converted to categorical, and (3) Non-numerical features (i.e., protected attributes such as sex, race, and age, and other demographics) were converted to numerical and coded to binary outcomes where 0 represented the under-represented/marginalized [1] group and 1 represented the over-represented/privileged group. The class outcomes were coded similarly, with 0 being negative and 1 being positive outcomes.

Table 1: Data sets descriptions including their domain usage and protected attributes.

| data set | Domain | Protected Attribute |
|---|---|---|
| Adult Census (ADU, 1994) | U.S. census information from 1994 to predict personal income | Sex, Race |
| Bank Marketing (BAN, 2017) | Marketing data of a Portuguese bank to predict term deposit | Age |
| COMPAS (Angwin et al., 2016) | Criminal history of defendants to predict re-offending | Sex, Race |
| German Credit (GER, 2000) | Personal information to predict good or bad credit | Sex |
| Communities (COM, 2009) | Law enforcement information to predict violent crimes | RacePctWhite |

## 5.2 ALGORITHM

Stealth has 7 main parts, as shown in Figure 2.

1. **Data Preparation:** The available data is divided 40:40:20 into a Parent:Surrogate:Test split. The Parent set is used to build the PARENT model (which may have hidden malicious model behavior).

---

[1]To clarify under-represented is the most common case in ML fairness problems because typically the marginalized group is the underrepresented or smaller population within the data set. However, I prefer to use marginalized because it can be extended to the COMPAS data set. The COMPAS data set is different because, in the case of racialized policing in the U.S., the marginalized group is the over-represented population.

2. **Baseline generation:** By training the PARENT using 40% training data and evaluating with the 20% test data, Alvarez & Menzies (2023) collected the *PARENT's baseline performance values* on the performance and fairness metrics. To create a default baseline, 80% of the data is used to train a full model.

3. **Recursive Bi-Clustering:** To downsample the SURROGATE training set into a representative subset, they used a recursive bi-clustering technique, Fastmap, which finds clusters of size $\sqrt{N}$, where N is the total amount of data points in the training set. One random example per leaf was used to build the SURROGATE downsampled subset. Alvarez & Menzies (2023) reference Chen et al. (2018), who reported Fastmap is fast and useful for finding a good representative subset of examples across data.

4. **Parent label probing:** To label the SURROGATE training set, Alvarez & Menzies (2023) probed the PARENT for labels, enabling the SURROGATE to mimic the same global behavior. The PARENT classifies every value in the SURROGATE's training set, and its classifications become the assigned label for those values instead of the original observed label. The SURROGATE models aim to "copy" PARENT behavior by extracting the labels by probing PARENT.

5. **Surrogate Generation:** The downsampled training set with labels from the PARENT is used to train a random forest to build a global SURROGATE model.

6. **Performance Evaluation:** Using the 20% test set, the evaluation metric scores for both the SURROGATE and PARENT are generated. These values are computed 20 times with reproducible random seeds, and all 9 metrics between the PARENT and SURROGATE are statistically compared with Scott-Knott's Statistical Test. The performance evaluation is used to measure how well the SURROGATE performs compared to the PARENT.

7. **Local Explanation Generation:** LIME reports the most influential features of both models. The top-ranked feature sets were compared using the Jaccard Index/Coefficient. The local evaluation explains the features of each model that influenced the specific outcomes in the test set.

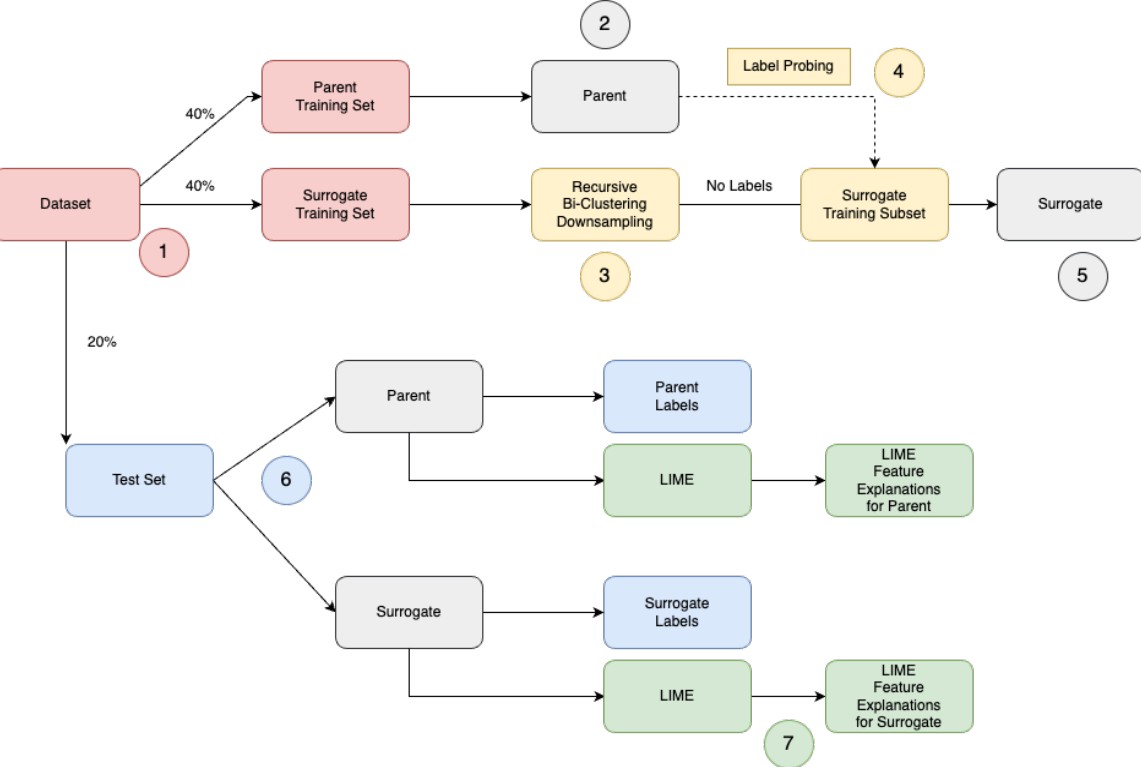

Figure 2: Stealth's full pipeline flowchart for surrogate generation and evaluation.

### 5.3 PARENT MODEL & SURROGATE GENERATION

After reviewing the pre-processing methods, we examined Stealth's surrogate generation. We looked into the code, and the Scikit-Learn train/test split method was stratified to ensure similar representation distributions in both training sets and the testing set. However, when we gathered the protected attribute representations in the downsampled set and the training sets, we found a **critical** problem.

There were significant class imbalances in the surrogate training set retained from the baseline (the model trained on 80% data set) shown in Figure 3. Specifically, when looking at the German data set, we can see that the privileged classes received the positive outcome 10x more than the marginalized class, and while the negative outcome is higher for the privileged, this percentage is proportional to the overrepresentation of the privileged class in the data set. The ratios between the baseline and the surrogate and the parent and the surrogate training subset are visually represented in Figure 3 and Figure 4.

The recursive bi-clustering method reduces the full training set into a *representative* subset, allowing the surrogate to copy the global behavior of the parent with 3-10% of the training set. This extremely limited use of data shown in Table 2 is one of Stealth's novel contributions. However, the imbalanced subgroup representation remains in the surrogate training set data distributions. This result highlights the accuracy of the downsampling technique but exposes the imbalanced and, therefore, unfair data distribution training the surrogate. Alvarez & Menzies' reported fairness intervention claim was misleading. Chakraborty et al. (2021) note the dangers of imbalanced demographic representation and how this unbalance leads to bias within the model's outcomes. With such extreme imbalance within the subsets, the surrogates had to present bias within its outcomes.

The Jaccard Coefficient evaluation of the parent and surrogate's influencing features, as reported by LIME, showed a 60% similarity for the parent and surrogate's top-ranked influencing features. The surrogates do perform just as well as the parent model despite their extreme imbalances. **The crucial point we make is that the surrogates do not improve the fairness properties of the parent model through a representative, downsampled subset because the surrogate mimics the outcomes of the parent model.**

This result revealed a flaw in the commonly held assumption that "more data" implies better results. When it comes to discriminatory/biased ML, if the original model or data is biased, the more data used only further reinforces said bias. Dressel & Farid (2018) supports this insight, which found that COMPAS models trained on 2 features performed just as well as models trained on all the features. A promising area for future work would be to investigate why Stealth's performance is statistically ranked fairer than its parent when the surrogate maintains the same unfair data imbalance.

Table 2: Sizes of the baseline (trained on 80% of the entire data set), parent, and surrogate model training sets.

| data set | Baseline | Parent | Surrogate | Surrogate % |
|---|---|---|---|---|
| Adult Census | 45,522 | 12,604 | 128 | 10% |
| Bank | 30,488 | 12,195 | 128 | 10% |
| COMPAS | 6,172 | 2468 | 64 | 3% |
| Communities | 123 | 49 | 16 | 33% |
| German | 1,000 | 400 | 32 | 8% |

## 6 RESULTS & DISCUSSION

While Alvarez & Menzies (2023) Scott Knott's Statistical Test analysis implied that Stealth was fairer than its parent model, further investigation revealed extremely imbalanced training sets. Stealth is presented as a "fairer" model generation, which also prevents malicious attacks. While the motivations behind this study are well-intentioned, this is an example of a model that does not fully achieve what the authors suggest it can do. After performing our procedural fairness audit and looking further into its subset generation, we find that there is still bias through data set imbalance. This finding exposes Stealth's claim of fairness as misguided and an example of how evaluation metrics are not the best way to determine "improved fairness." The main source of bias comes from

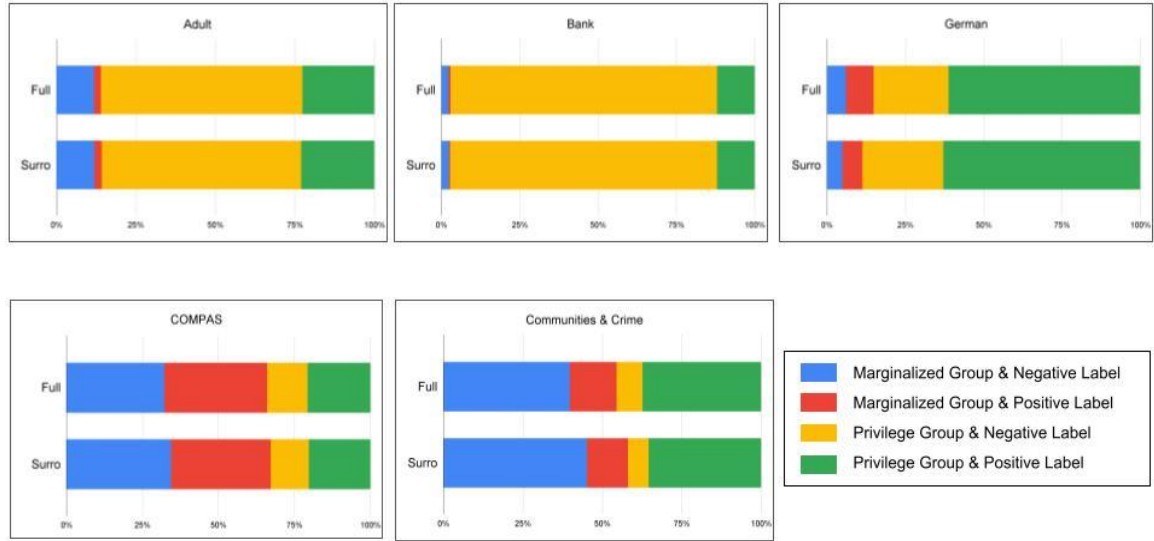

Figure 3: The Baseline model's positive and negative labels are separated by the marginalized and privileged protected attributes and compared to the downsampled Surrogate's percentages. A similar label imbalance in the Baseline is maintained in the Surrogate.

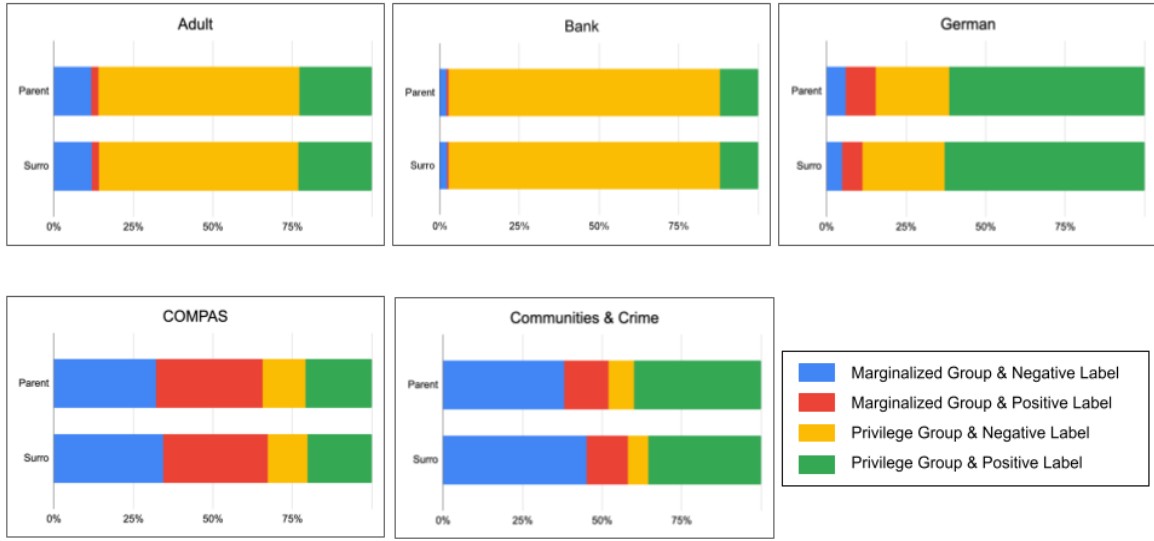

Figure 4: The Parent model's positive and negative labels are separated by the marginalized and privileged protected attributes and compared to the downsampled Surrogate's percentages. A similar label imbalance from the Parent is maintained in the Surrogate.

looking further into the data representation ratios within the training, test, and down-sampled sets shown in Figures 3 and 4.

In a more positive light, the contribution of Stealth's work is the impressive application of their recursive bi-clustering downsampling technique to surrogate generation. Stealth's downsampling technique did create an accurate/representative subset, as supported by the data representation ratios seen in Figures 3 and 4. Also, the surrogate models were successful at duplicating the parent's global behavior, as supported by Alvarez & Menzies (2023) LIME explanations and statistical analysis with the Jaccard Coefficient score.

With the same data imbalance as the baseline and parent set, the surrogates performed statistically similar to the parent model. This result contributes to a further understanding of the root of bias and how little data can bias a model. With a lot of attention focused on deep learning and big data modeling, we report that if models with a median of 10% of a data set are able to perform comparatively and with similar bias, the impact of deep learning models should be heavily reviewed. Alvarez & Menzies (2023) experimented with random forests, which Molnar remarks are "strong learners," but there are other learners that future work can investigate (Molnar, 2023).

## 7 CONCLUSION

This study developed a procedural justice audit and used the audit to assess an XAI case study, Stealth. The procedural justice audit exposed the imbalanced training sets and made us question Alvarez & Menzies' claims to improve fairness. Our procedural fairness audit of Stealth is an example of a *meta-audit* where an XAI, auditing tool is being further evaluated. Generalized fairness interventions oversimplify the nuances of bias and discrimination within AI models. Thus, fairness and bias mitigation properties must be contextualized and procedurally evaluated per model. While Stealth fails to meet its fairness claims, it does use a representative downsampling technique to create viable global surrogates that avoid evaluation detection by deceptive AI.

We caution future researchers against only using metric thresholds as fairness evaluations and to re-focus on procedural fairness. Whereas other XAI have a general assumption that the model-owners are honest. There is a need for better XAI tools, and as deceptive AI is gaining traction, there's an increasing need for evaluation tools to give a sense of security that these models aren't intrinsically or intentionally biased. This is especially important as more ML services (Marketplace, 2019; Ribeiro et al., 2015) and models from the cloud are being marketed and widely accepted, as we've seen with OpenAI's ChatGPT. This proposed *procedural fairness audit* is an example of an iterative XAI audit and advocates for a more comprehensive implementation of "procedural fairness," which emphasizes justice beyond "fair outcomes." The results of the audit contribute to the intersectional work of IML, XAI, Trustworthy ML, and security (deception detection).

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

## A  APPENDIX

You may include other additional sections here.

