# OpenReview forum: "XAI Procedural Fairness Auditing Framework: avoid misguided outcomes by refocusing on fairness properties"
_ICLR.cc/2024/Conference — ICLR 2024 Conference Withdrawn Submission_

### Official Review · Reviewer_wYqY · 2023-10-12

**Soundness:** 2 fair
**Presentation:** 1 poor
**Contribution:** 1 poor
**Rating:** 1
**Confidence:** 3

**Summary:**

Authors present the results of a new procedural fairness auditing framework for explainable AI tools by evaluating Stealth.
Authors audited Stealth’s decision-making process outside of its notable performance outcomes.
The procedural fairness audit reports that Stealth’s global surrogate models are impressive and a successful application of recursive bi-clustering for representative data downsampling.

**Strengths:**

The idea of procedural fairness, even if not completely clear, may be interesting.

**Weaknesses:**

Novelty of the paper is limited
The technical contribution of the paper is limited.
There is heavy repetition of know material.
Paper is hard to grasp and read.

**Questions:**

It is not clear
- the contribution of the paper
- the novelty of the paper
- the result that do not prove a point

---

### Official Review · Reviewer_VbpT · 2023-11-01

**Soundness:** 2 fair
**Presentation:** 3 good
**Contribution:** 1 poor
**Rating:** 3
**Confidence:** 3

**Summary:**

This study conducts a procedural fairness auditing for Stealth's decision-making process. The findings indicate Stealth has several fairness issue.

**Strengths:**

The research question is interesting. Auditing the fairness of the decision-making process is crucial to ensure procedural fairness of ML decision-making.

**Weaknesses:**

The scalability of the proposed framework is a concern, as all the current conclusion and design is only from one case study of Stealth.

**Questions:**

What motivates the 11 questions in your framework?
How this proposed procedural fairness audit can be used for other AI-supported decision-making systems?

---

### Official Review · Reviewer_g6MX · 2023-11-02

**Soundness:** 2 fair
**Presentation:** 2 fair
**Contribution:** 2 fair
**Rating:** 5
**Confidence:** 4

**Summary:**

The article discusses the application of machine learning (ML) auditing, particularly focusing on procedural fairness in explainable AI (XAI) methods. Procedural fairness emphasizes the fairness of the decision-making process rather than the outcomes. The authors introduce a procedural fairness auditing framework applied to an XAI tool called Stealth, which integrates a novel global surrogate model with local explanations from LIME, aiming to be undetectable by deceptive models. The audit assesses Stealth's decision-making process independently from its performance outcomes. Findings show that while Stealth's global models and data downsampling techniques are effective, there are inherent biases in its training data. This leads to a critical examination of Stealth's fairness claims, which appeared to be skewed towards "fairer outcomes" rather than actual fair processes. The framework outlined in the study serves as a guide for ensuring procedural fairness in ML decision-making.

**Strengths:**

1. This paper provides a novel application of procedural fairness through an "auditor-in-the-loop" auditing framework. The iterative audit process allows for a more comprehensive evaluation of an AI system's decision-making process and fairness properties.
2. The framework is a significant step forward in ML auditing because it focuses on the fairness of the decision-making process itself, rather than just the outcomes.
3. The case study on Stealth exposes how its claims of "fairness" were misguided, demonstrating the importance of looking beyond metrics to truly evaluate procedural fairness.

**Weaknesses:**

1. The rationale for choosing Stealth as the tool to study procedural fairness could be clarified.
2. The paper would benefit from being more self-contained. Currently, readers need to consult the cited Alvarez & Menzies (2023) paper and find the reference to Stealth in the abstract. Providing more context about Stealth (e.g., its Github link) within this paper would make it more accessible for readers.
3. The audit relies on some evaluation results reported in the original Stealth paper without full independent verification. Reproducing those experiments could further validate the analysis.

**Questions:**

Please refer to the Weakness section

---

### Official Review · Reviewer_VwFX · 2023-11-02

**Soundness:** 2 fair
**Presentation:** 2 fair
**Contribution:** 2 fair
**Rating:** 3
**Confidence:** 4

**Summary:**

The paper focuses on one particular explainable AI method, Stealth, and aims to present auditing results for procedural fairness. The paper is more of a meta analysis, instead of a detailed technical treatment. Empirical results are also presented.

**Strengths:**

The paper attempts to draw connection between two important topics: XAI and procedural fairness.

**Weaknesses:**

The weakness of the paper comes from the lack of organization of the material, as well as the unclear presentation of the problem of interest. The paper is not easy to follow. Multiple different topics are discussed at the same time. Readers can benefit from a more organized presentation, and a more developed solution towards the problem of interest.

**Questions:**

Q: what is the intended connection between XAI and fairness that authors would like to draw?

Section 3 presents the motivation of case study. Section 4 lists multiple questions in procedural fairness audits. Section 5 introduces in detail the Stealth approach for XAI. Readers can benefit from a more organized way of presenting/investigating the connection between XAI and fairness. The current version invites more questions than answers, and the takeaway message is not clear.